## TOOLS AND RESOURCES

# Toggle-Untoggle – a cell segmentation tool with an interactive user verification interface

Nina Grishchenko[1], Margarita Byrsan[2], Matthew Craig Drummond-Stoyles[3] and Michael F. Olson[1,2,4,*]

## ABSTRACT

Accurate cell segmentation is an essential step in the quantitative analysis of fluorescence microscopy images. Pre-trained deep learning models for automatic cell segmentation such as those offered by Cellpose perform well across a variety of biological datasets but might still introduce segmentation errors. Although training custom models can improve accuracy, it often requires programming expertise and significant time, limiting the accessibility of automatic cell segmentation for many wet lab researchers. To address this gap, we developed 'Toggle-Untoggle', a standalone desktop application that enables intuitive, code-free quality control of automated cell segmentation. Our tool integrates the latest Cellpose 'cyto3' model, known for its robust performance across diverse cell types, while also supporting the 'nuclei' model and user-specified custom models to provide flexibility for a range of segmentation tasks. Through a user-friendly graphical interface, users can interactively toggle individual segmented cells on or off, merge or draw cell masks, and export morphological features and cell outlines for downstream analysis. Here, we demonstrate the utility of Toggle-Untoggle in enabling accurate, efficient single-cell analysis on real-world fluorescence microscopy data, with no coding skills required.

KEY WORDS: Image analysis, Cell morphology, Fluorescence microscopy, Cytoskeleton

## INTRODUCTION

Accurate cell segmentation is a necessary step for robust quantitative analysis of fluorescence microscopy images, enabling the identification of individual cells and the extraction of morphological features, such as those related to size, shape and intensity, which are essential for studying cell behaviors, phenotypic variation and treatment responses (Li et al., 2013). These morphological features are often used in downstream applications, such as clustering and classification, where segmentation quality directly impacts the reliability of the results (Ahmadzadeh et al., 2017; Yao et al., 2019)

Deep learning approaches, particularly convolutional neural networks (CNNs) such as U-Net, have significantly advanced segmentation accuracy in bioimage analysis (Ronneberger et al., 2015). A U-Net style architecture called Cellpose has gained popularity for its generalist pre-trained models that perform well across a range of imaging conditions and cell types (Stringer et al., 2021). The latest Cellpose version introduced the 'cyto3' model, which improved segmentation performance while eliminating the need for user-specific model training, making it more accessible to users without deep learning expertise (Stringer and Pachitariu, 2025).

Despite these advantages, pre-trained models are not always well-suited to every dataset. In cases where cell shapes, textures or imaging conditions markedly differ from the training data, segmentation errors can occur, including missing cells, over-segmentation or poor boundary detection, each of which could compromise the accuracy and robustness of extracted morphological data (Chen and Murphy, 2023). Manual correction is often necessary, which can be slow and difficult when processing large volumes of images.

To address this challenge, we developed 'Toggle-Untoggle', a standalone desktop application (app) with a graphical user interface (GUI) that integrates automated segmentation using a user-friendly verification tool with streamlined extraction of morphological parameters. Although Toggle-Untoggle is compatible with cyto3, 'nuclei' and custom Cellpose models, our focus in this study is on the cyto3 model, due to its broad applicability to cytoplasmic and whole-cell segmentation tasks (Stringer et al., 2021; Stringer and Pachitariu, 2025).

The app features an interactive interface with a user input panel (Fig. 1A), built-in guidance, and troubleshooting tips to help minimize segmentation errors. Designed for accessibility, it requires no coding expertise, making it easy for users of all backgrounds to perform high-quality single-cell image analysis. Images in TIFF format can be automatically loaded from a specified directory for batch processing. Following segmentation, the app extracts standard morphological features for each detected cell and enables users to review and refine results (Fig. 1B). Individual segmented cells can be toggled off to exclude them from downstream analysis, with their positions retained for optional reactivation. Beyond toggling, users can merge adjacent masks or draw new cell boundaries to add missing cells. This interactive system provides an efficient way to enhance segmentation quality without the need for extensive manual annotations or the time-consuming identification of outliers in downloaded datasets. In addition, the app enables users to export results in formats compatible with widely used analysis platforms (Fig. 1B). A .csv file containing single-cell morphological parameters, as well as segmentation masks in .roi format, can be saved for further analysis in external tools, such as Fiji or QuPath (Bankhead et al., 2017; Schindelin et al., 2012). These features increase the utility of Toggle-Untoggle for users who require more advanced visualization or analysis beyond the app itself.

Here, we present the design and functionality of the Toggle-Untoggle app, demonstrate its performance on an image dataset and highlight its utility in enabling accurate, scalable and user-friendly

[1]Department of Chemistry and Biology, Toronto Metropolitan University, Toronto, ON M5B 2K3, Canada. [2]Biomedical Engineering Program, Toronto Metropolitan University, Toronto, ON M5B 2K3, Canada. [3]Department of Mechanical and Industrial Engineering, University of Toronto, Toronto ON M5S 1A1, Canada. [4]Department of Pharmacology and Toxicology, University of Toronto, Toronto ON M5S 1A1, Canada.

*Author for correspondence (michael.olson@torontomu.ca)

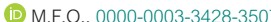 M.F.O., 0000-0003-3428-3507

## A

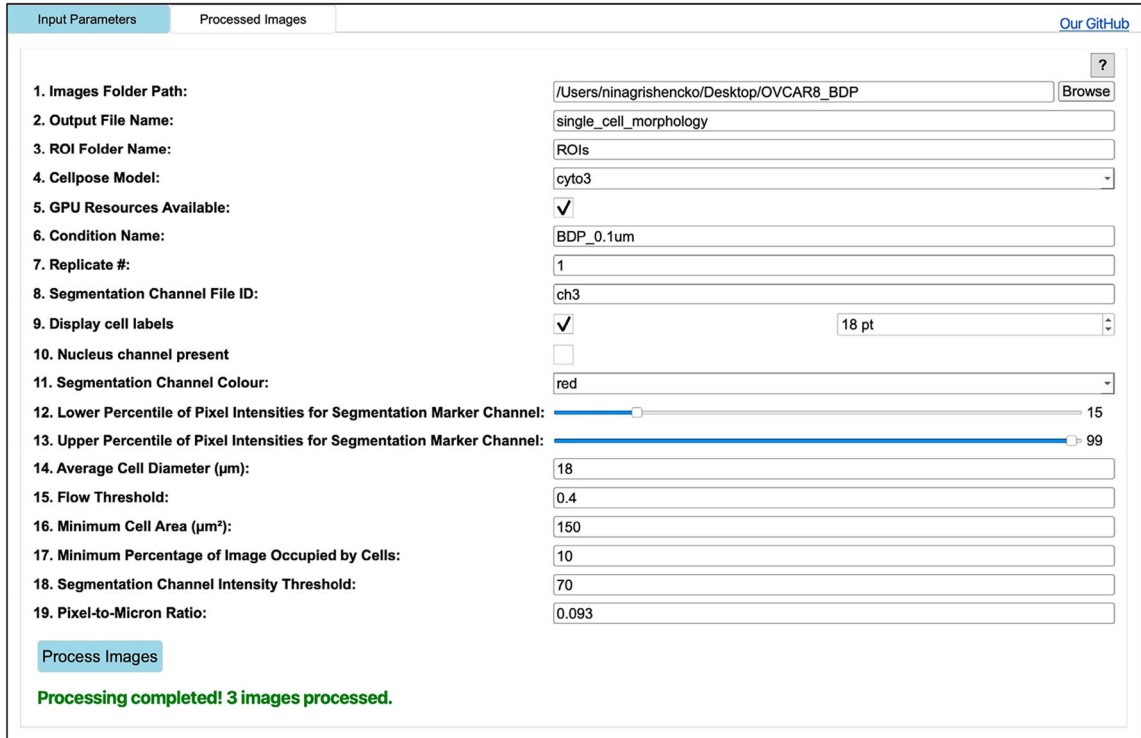

## B

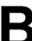
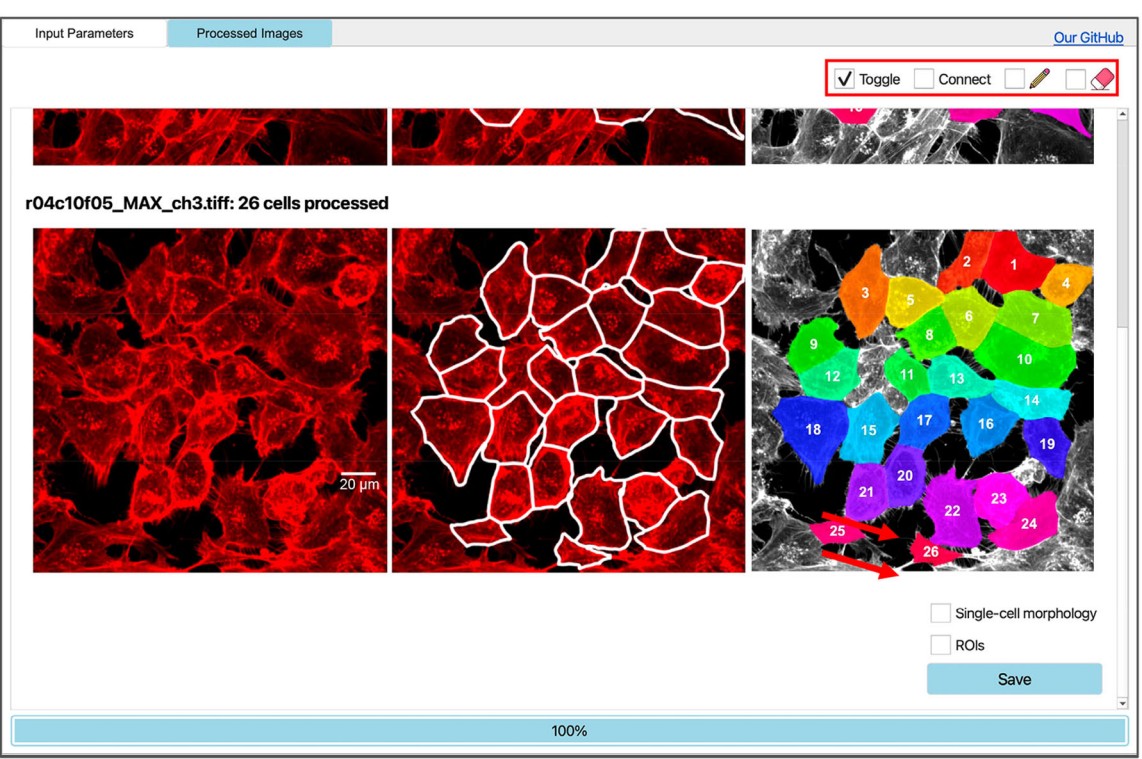

**Fig. 1. Toggle-Untoggle graphical user interface.** (A) An overview of the Input Parameter screen that allows users to upload image datasets and vary conditions for optimal segmentation. (B) Images of parental OVCAR-8 cells were used as an example segmentation. Fluorescently labelled phalloidin was used to visualize F-actin, which served as the marker for cell segmentation. Inside the red box in the top right corner, users can select from various modes – Toggle, Connect, Draw and Erase – by checking the corresponding boxes. Red arrows in the bottom right corner highlight the options located above the 'Save' button, allowing users to export a .csv file containing morphological parameters and/or region of interest (ROI) outlines for individual cells of interest. Scale bar: 20 μm.

single-cell analysis of fluorescence microscopy images of mammalian cells.

## RESULTS

A custom GUI app was developed with PyQt6 to facilitate automated cell segmentation, morphological feature extraction and user-guided verification. The app supports custom Cellpose models and includes the pre-trained cyto3 and nuclei models that can be selected from the model dropdown menu, enabling immediate use without any custom training (Stringer et al., 2021; Stringer and Pachitariu, 2025). Upon launch, users select a directory containing input images, which are automatically loaded and processed in batch mode (Fig. 1A). The interface includes adjustable input fields for key parameters, such as the channels to be analyzed, display settings to enhance false-color images for easier post-segmentation verification, estimated cell diameter and flow threshold – the maximum allowed error per mask (Stringer et al., 2021). Additional parameters include a minimum cell area to eliminate debris and under-segmented cells, as well as intensity thresholds to filter out empty images and objects lacking a nucleus. These are supplemented with on-screen instructions to assist users in optimizing segmentation performance and troubleshooting if required (Fig. 1A).

For each image, Cellpose is applied to generate cell outlines and segmentation masks based on the chosen cytoplasmic marker. These masks are overlaid on the original image and displayed in the GUI for review (Fig. 1B). After processing all images in the directory, or only a subset using the 'stop processing' button, users can review and modify the segmentation results. For images acquired at lower magnifications (e.g. 20× or below) or containing a large number of small cells, zooming and panning are available to facilitate detailed inspection. Exportable .csv files include the morphological parameter data for each cell. Regions of interest (ROIs) for the segmented cells can also be saved for downstream analysis in external tools, if applicable (Fig. 1B). All output files are saved to the folder containing the input images, using a custom name based on the analysis settings or user input.

As Toggle-Untoggle introduces additional parameters beyond the standard ones used to fine-tune Cellpose model performance, we compared the segmentation results of the pre-trained Cellpose cyto3 model and Toggle-Untoggle against manually annotated ground truth using IoU scores (Fig. 2). Although the difference was not statistically significant, Toggle-Untoggle trended towards slightly better performance on a test subset of images containing MDA-MB-231 cells stained with phalloidin, suggesting that there might be benefits of applying this tool to specific cell lines or experimental conditions.

Given that pre-trained models might not always yield perfect results, particularly in the presence of cell clusters or artifacts, the app includes the interactive Toggle-Untoggle feature, along with options to merge neighboring cells and draw missing ones. As shown in the example of MDA-MB-231 breast cancer cells that were segmented based on expression of a GFP marker, even after troubleshooting with all the available parameters in the user input panel on the first tab of the app, incorrectly segmented or missing cells could still occur (Fig. 3). This can cause users to doubt the accuracy of the segmentation and prompt them to correct mis-segmented objects. Toggle-Untoggle provides a straightforward solution to this commonly encountered problem. The interactive tool built into the GUI allows users to get rid of incorrectly segmented cells by simply clicking on them, which increases the opacity of their masks and ensures that these cells are not included in

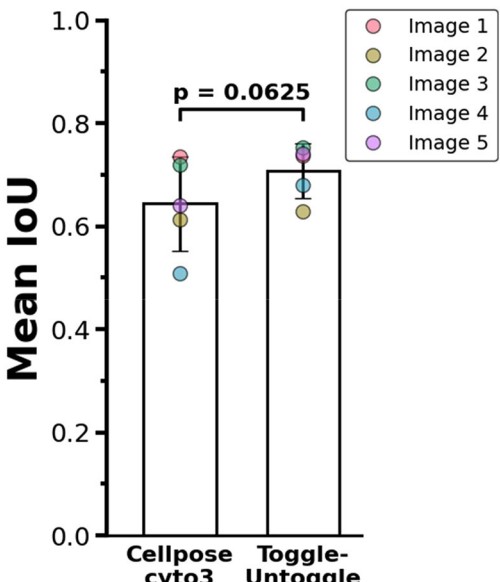

**Fig. 2. Segmentation performance comparison of Toggle-Untoggle and Cellpose cyto3 against manual ground truth.** MDA-MB-231 cells were segmented from single-channel fluorescence images stained for F-actin, in the absence of nuclear labeling. Manual ground truth annotations were created in Fiji. The Intersection over Union (IoU) metric was used to evaluate and compare the segmentation performance of the Cellpose cyto3 model and the Toggle-Untoggle tool, both applied in a fully automated, non-interactive mode. A total of five independent images were used for this comparison. Means±s.e.m. Statistical analysis was performed using the Wilcoxon signed-rank test, with a $P<0.05$ considered to be statistically significant.

the exported dataset or saved ROIs (Fig. 3A). Users also have the option to restore 'toggled' masks for each image by clicking on their opaque version, at which point the cell mask will revert to the original settings and be returned in the exportable dataset and ROIs, should it be later determined that their exclusion was an error.

In addition to toggling, the app supports merging and drawing functionalities. As shown in Fig. 3B, neighboring over-segmented cells can be merged by drawing a line across their shared boundary, while Fig. 3C illustrates how missing cells can be manually added using the drawing tool integrated into the GUI. Both actions are fully reversible, allowing users to freely modify segmentations without risk of permanent data loss.

Users also have the option to save two .csv files – one containing verified segmented cells, and another containing cells that were toggled off by the user. Both files include the morphological parameter data for each cell. The possibility of exporting two separate files containing either the accepted or the removed cells could allow for refinement of automated segmentation algorithms by incorporating training for acceptable and unacceptable cell morphologies. Verifications and adjustments can be performed across all images and cells displayed in the GUI, which is then implemented in the exported data.

Following segmentation and optional verification, single-cell morphological parameters are automatically calculated using the scikit-image library. Some of the extracted features include area, perimeter, eccentricity, solidity, major and minor axis lengths and mean fluorescence intensity. These parameters provide quantitative descriptions of cell shape and signal, enabling detailed comparisons between different experimental conditions. Here, we demonstrate how the functionality of Toggle-Untoggle was used to compare the

## MDA MB 231

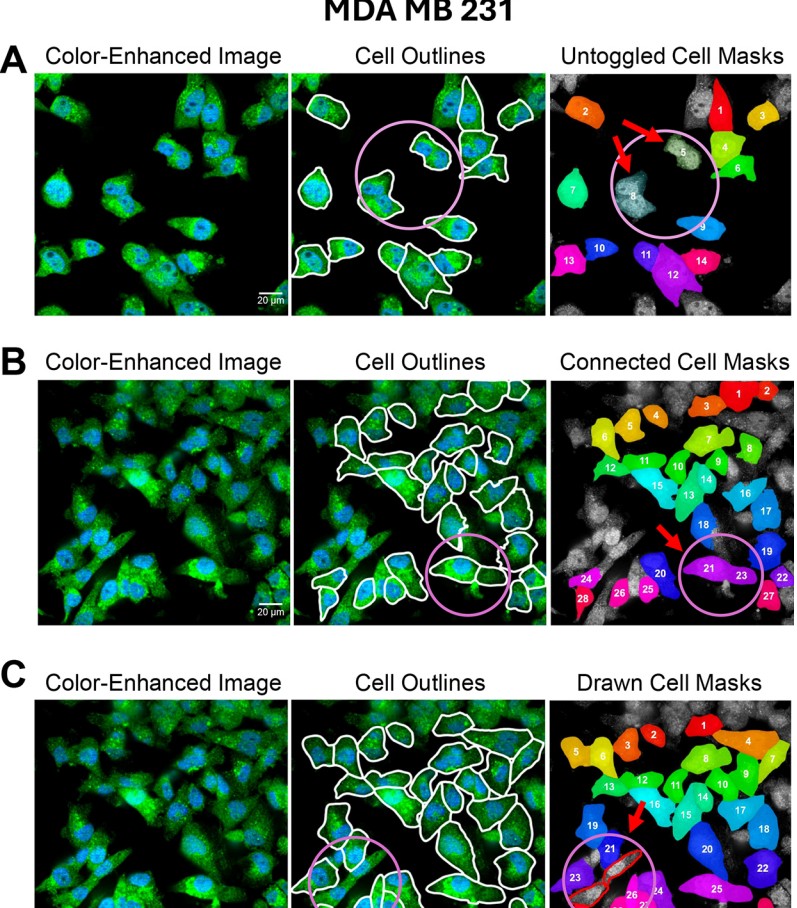

**Fig. 3. Interactive segmentation refinement in Toggle-Untoggle – de-selection, merging and manual annotation.** MDA-MB-231 cells were segmented based on GFP expression. DAPI channel images were used to refine the segmentation by excluding cells lacking nuclei. (A) Segmentation masks can be toggled on or off by the user. De-selected masks are rendered opaque and excluded from the final dataset. (B) Segmented regions initially identified as separate objects can be merged to be counted as a single cell. (C) Users can manually draw cell outlines as needed. In all panels, relevant objects are enclosed within a purple circle and indicated by red arrows. Scale bars: 20 µm.

morphology of parental and cisplatin-resistant OVCAR8 cells, which appear to differ in size from a qualitative perspective. We segmented A2780 (Fig. 4A) and A2780CisR (Fig. 4B) cells based on their F-actin structures that were visualized with fluorescent phalloidin. Using the quantitative single-cell morphological data acquired with Toggle-Untoggle, we then performed statistical

## A2780

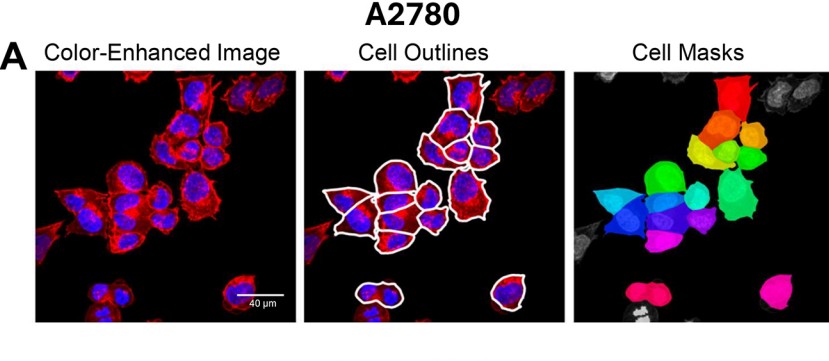

## A2780CisR

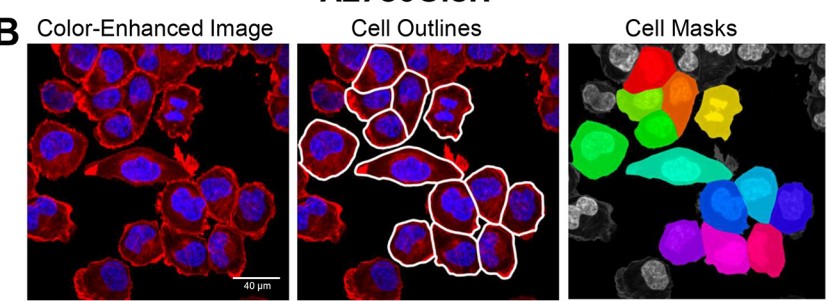

**Fig. 4. Segmentation of parental and cisplatin-resistant A2780 cell lines.** (A) Parental A2780 and (B) cisplatin-resistant A2780CisR cell lines were segmented based on phalloidin staining of F-actin structures. DAPI channel images were used to refine the segmentation output by excluding cells lacking nuclei. Images were analyzed using the Toggle-Untoggle GUI to extract single-cell morphological parameters and regions of interest. Scale bars: 40 µm.

analyses to compare the appearance of parental and cisplatin-resistant A2780 cells. Cell area (Fig. 5A), bounding box area (Fig. 5B), and perimeter (Fig. 5C) were each statistically significantly different between parental A2780 and cisplatin-resistant A2780CisR ovarian cancer cells.

To demonstrate the feasibility of downstream analysis using the cell segmentation masks produced by Toggle-Untoggle, the ROI masks generated by the app were saved and imported into the third-party platform QuPath (Bankhead et al., 2017) to extract Haralick texture features, enabling an evaluation of differences in actin texture between the parental and cisplatin-resistant A2780 cell populations. Haralick features can be used to quantify textural properties in images by capturing patterns of pixel intensity variation, which can reflect underlying biological structures, such as the organization of the actin cytoskeleton (Hamilton, 2009; Hui et al., 2023) In this study, we determined that specific Haralick features were statistically significantly different between parental A2780 and cisplatin-resistant A2780CisR ovarian cancer cells, including contrast, which measures the extent of local variations present in the image (Fig. 5D); and angular second momentum, which reflects how uniform or consistent the intensity patterns are throughout the image (Fig. 5E) (Haralick et al., 1973; Soh and Tsatsoulis, 1999). In addition, we used the FibrilTool Fiji plugin to evaluate anisotropy in order to compare the degree of alignment of actin filaments within the parental A2780 and cisplatin-resistant A2780CisR ovarian cancer cells (Boudaoud et al., 2014; Schindelin et al., 2012). Together, these metrics provide complementary information about cytoskeleton organization, offering insights into structural changes associated with drug resistance.

## DISCUSSION

Accurate segmentation is a key requirement for reliable single-cell microscopy image analysis (Li et al., 2013). Although generalist models such as Cellpose (Stringer et al., 2021) have been developed to work across a variety of cell types without the need for users to train their own models, the results can still be suboptimal and require manual verification. Here, we introduce Toggle-Untoggle, a tool designed to assist non-specialist researchers in analyzing fluorescence microscopy images at the single-cell level.

In comparison to existing tools, our approach offers unique advantages tailored to wet-lab workflows. Although Cellpose itself

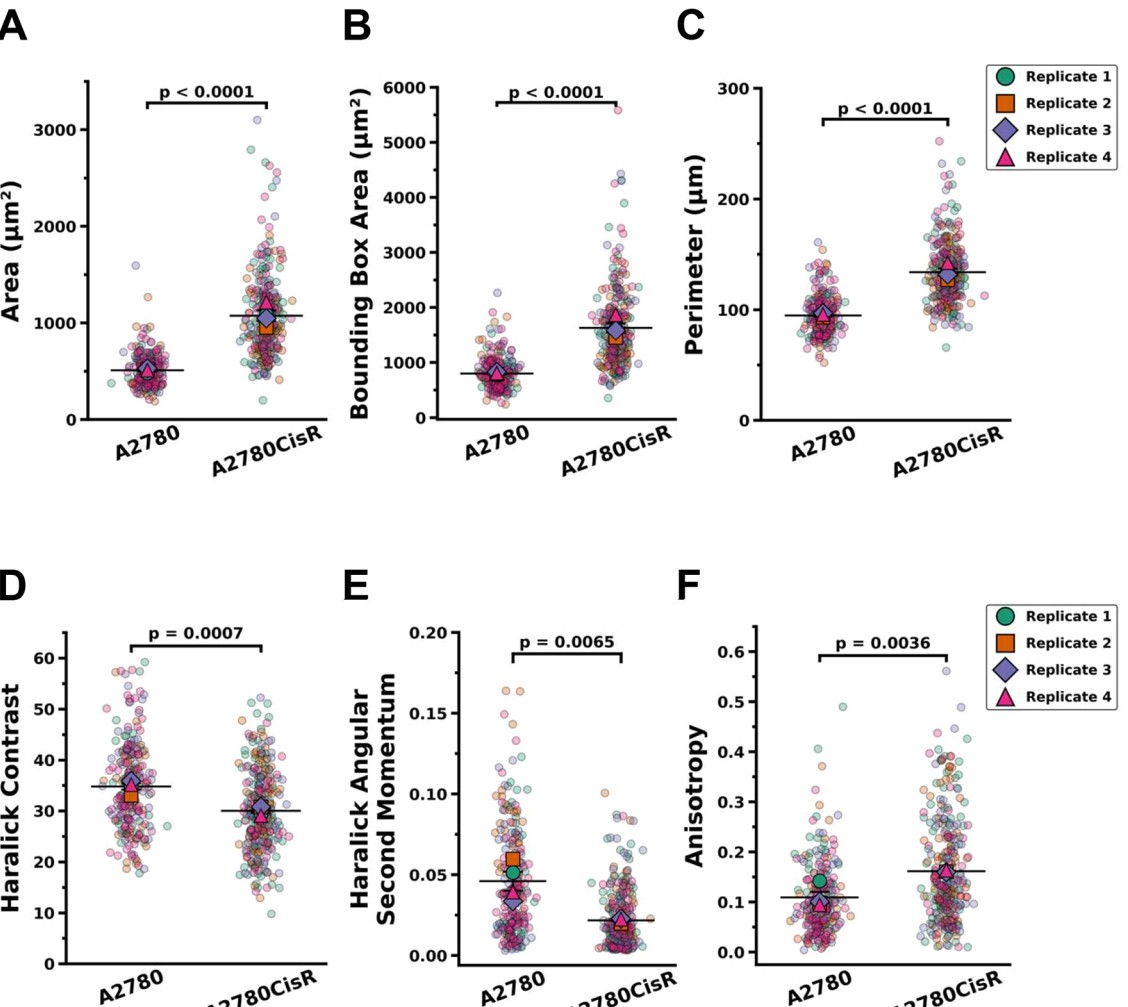

**Fig. 5. Comparative morphological analysis of parental and cisplatin-resistant A2780 cells.** Cell segmentation and morphological feature extraction for parental A2780 and cisplatin-resistant A2780CisR cells were performed using Toggle-Untoggle, enabling quantification of (A) cell area, (B) cell bounding box area, and (C) cell perimeter. ROIs were exported from Toggle-Untoggle and further analyzed in QuPath and Fiji to extract texture and shape descriptors. Specifically, QuPath was used to compute Haralick texture features (D), contrast and (E) angular second moment, while Fiji was used to calculate (F) cell anisotropy. Statistical comparisons between parental and resistant cell lines were conducted using a two-tailed unpaired Student's $t$-test, with $P<0.05$ considered to be statistically significant. Data represent four replicates ($n=4$) per condition; horizonal lines indicates the means.

includes a GUI, it is mainly intended for parameter optimization and does not support bulk image analysis (Stringer and Pachitariu, 2025). Although comprehensive platforms, such as Fiji (Schindelin et al., 2012) and QuPath (Bankhead et al., 2017), provide plugins compatible with Cellpose for batch processing, they often require at least some degree of coding proficiency and do not offer interactive options for verifying segmentation masks, a key feature of Toggle-Untoggle.

Nevertheless, our method has certain limitations. Although manual verification of segmentation masks can enhance accuracy, it can also introduce some level of subjective bias if not systematically approached. To minimize user-dependent variability within a single experiment, we recommend that one analyst conduct the review. Furthermore, because the software requires periodic user intervention, it is better suited to low-throughput analyses rather than to high-content, large-scale screening applications. Performance constraints should also be noted – although Toggle-Untoggle can leverage the embedded GPU available on M1–M3 processors from Apple, its partial reliance on CPU resources might still limit performance on large datasets. On other operating systems, access to additional GPU resources is strongly recommended to achieve optimal user experience.

## MATERIALS AND METHODS
### Cell culture
Human ovarian endometrioid adenocarcinoma cells (A2780) and a cisplatin-resistant derivative (A2780CisR), obtained from Dr Robert Rottapel (Princess Margaret Cancer Centre, Toronto, Canada), were cultured in RPMI 1640 medium (Sigma-Aldrich, R8758) supplemented with 10% fetal bovine serum (FBS, Thermo Fisher Scientific, 12483020), 100 U/ml penicillin and 100 µg/ml streptomycin (Gibco, 15140122). Triple-negative human breast cancer (TNBC) MDA-MB-231 cells (Caliper LifeScience) were cultured in Dulbecco's modified Eagle's medium (DMEM, Sigma-Aldrich, D5796) supplemented with 10% FBS, 100 U/ml penicillin and 100 µg/ml streptomycin. All cells were routinely tested for mycoplasma contamination, and were maintained at 37°C in a humidified incubator with 5% $CO_2$.

### Immunofluorescence microscopy
A2780 and A2780CisR cells were seeded at 5000 cells per well, and MDA-MB-231 cells at 12,000 cells per well, in PerkinElmer PhenoPlate 96-well CellCarrier Ultra plates. Cells were cultured for 24 h at 37°C with 5% $CO_2$. After three washes with PBS, cells were fixed with 30 µl/well of 4% paraformaldehyde (PFA) in PBS for 15 min at room temperature in the dark. Permeabilization was performed with 50 µl/well of 0.1% Triton X-100 in PBS for 5 min, followed by blocking with 50 µl/well of 3% (w/v) bovine serum albumin (BSA) in PBS for 1 h at room temperature.

A2780 (RRID:CVCL_0134) and A2780CisR (RRID:CVCL_H745) cells were stained with Rhodamine-conjugated phalloidin (1:200, Thermo Fisher Scientific) and 4′,6-diamidino-2-phenylindole (DAPI, 1:5000, Thermo Fisher Scientific) in 3% BSA for 1 h to label F-actin and nuclei, respectively. MDA-MB-231 (RRID:CVCL_0062) cells were stained with Rhodamine-conjugated phalloidin (1:200) and DAPI (1:5000) as above, along with an anti-GFP N-terminal rabbit polyclonal antibody (1:200, G1544, Sigma-Aldrich, RRID:AB_439,690) in 3% BSA for 1 h. After washing three times with PBS, cells were incubated with Alexa Fluor® 488-AffiniPure donkey anti-rabbit IgG secondary antibody (1:1000, A-21206, Thermo Fisher Scientific, RRID:AB_2535792) in 3% BSA for 1 h in the dark. Following three final PBS washes, cells were stored in 50 µl PBS at 4°C in the dark until imaging. High-content imaging was performed using an Opera Phoenix™ Plus high-content imaging system.

### Graphical user interface development workflow
Toggle-Untoggle was built with PyQt6 and packaged with PyInstaller. It uses the Cellpose algorithm for cell segmentation and scikit-image python package to extract morphological parameters of individual objects. The full code and first release of the app are available to view and download on our GitHub page at https://github.com/ninagris/Toggle-Untoggle.

### Data analysis and visualization
The Wilcoxon signed-rank test was used to compare IoU scores between Cellpose cyto3 model and Toggle-Untoggle. For morphological features, all statistical comparisons were performed using a two-tailed unpaired Student's $t$-test. A $P≤0.05$ was considered statistically significant. All statistical analyses were conducted using the scipy.stats package in Python. Data visualization was carried out using the Python library matplotlib. Haralick texture features were extracted and analyzed with QuPath (version 0.5.1). Anisotropy was calculated in Fiji (version 2.16.0/1.54p) using the FibrilTool plugin.

### Acknowledgements
A2780 and A2780CisR cells were a kind gift from Dr Robert Rottapel (Princess Margaret Cancer Centre, Toronto Canada).

### Competing interests
The authors declare no competing or financial interests.

### Author contributions
Conceptualization: N.G., M.F.O.; Data curation: N.G., M.B.; Formal analysis: N.G., M.B., M.F.O.; Funding acquisition: M.F.O.; Investigation: N.G., M.B.; Methodology: N.G., M.B., M.F.O.; Project administration: M.F.O.; Resources: M.F.O.; Software: N.G., M.C.D.-S.; Supervision: M.F.O.; Validation: M.F.O.; Visualization: N.G., M.F.O.; Writing – original draft: N.G., M.F.O.; Writing – review & editing: M.F.O.

### Funding
This research was supported by funding to M.F.O. from the Canadian Institutes of Health Research (PJT-169125), Natural Sciences and Engineering Research Council of Canada (RGPIN-2020-05388), and Canada Research Chairs Program (950-231665). Open Access funding provided by University of Toronto. Deposited in PMC for immediate release.

### Data and resource availability
Toggle-Untoggle is available at GitHub at https://github.com/ninagris/Toggle-Untoggle. All relevant data and details of resources can be found within the article.

### First Person
This article has an associated First Person interview with the first author of the paper.

### Peer review history
The peer review history is available online at https://journals.biologists.com/jcs/lookup/doi/10.1242/jcs.264154.reviewer-comments.pdf

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
