## [Peer Review File · Journal of Cell Science]

Toggle-Untoggle: A cell segmentation tool with an interactive user verification interface

Nina Grishchenko, Margarita Byrsan, Matthew Craig Drummond-Stoyles and Michael F. Olson

DOI: 10.1242/jcs.264154

Editor: Guillaume Jacquemet

Review timeline

Original submission:	20 May 2025
Editorial decision:	26 June 2025
First revision received:	25 August 2025
Editorial decision:	17 September 2025
Second revision received:	26 September 2025
Accepted:	1 October 2025

Original submission

First decision letter

MS ID#: jcs.264154

MS TITLE: Toggle-Untoggle: A cell segmentation tool with an interactive user verification interface

AUTHORS: Nina Grishchenko; Margarita Byrsan; Michael F Olson

ARTICLE TYPE: Tools and Resources

Dear Dr Olson,

We have now reached a decision on the above manuscript.

To see the reviewers' reports and a copy of this decision letter, please go to:

As you will see, the reviewers raise a number of substantial criticisms that prevent me from accepting the paper at this stage. They suggest, however, that a revised version might prove acceptable, if you can address their concerns.

I am also especially concerned about the lack of support for other OS than certain Mac. If you think that you can deal satisfactorily with the criticisms on revision, I would be pleased to see a revised manuscript. We would then return it to the reviewers.

Reviewer 1

SUMMARY OF THE ADVANCE MADE IN THIS PAPER AND ITS POTENTIAL SIGNIFICANCE TO THE FIELD

In this paper, "Toggle-Untoggle: A cell segmentation tool with an interactive user verification interface," Grishchenko et al. present a desktop tool offering a user-friendly interface for applying the Cyto3-based segmentation pipeline to cell images in batch. This tool performs automated segmentation on a series of images using the Cellpose cyto3 model, extracts shape and texture features, exports results in CSV format and ROIs. This manuscript also showcases a possible downstream analysis using QuPath.

The tool is well-suited for wet lab researchers with no coding experience who work with cell morphological profiling. It shows potential for high-throughput and reproducible image analysis. Overall, the manuscript is well written. Toggle-Untoggle is nicely designed and provides an easy go-to tool for cell segmentation and morphological studies. Although the tool is already usable, it would benefit from expansion to allow more flexibility.

This review focuses mainly on evaluating the usability of Toggle-Untoggle and providing suggestions for extensions for broader applicability.

SUGGESTIONS TO AUTHORS

1. Graphical user interface

- * The GUI currently launches in full-screen mode by default. It would be better to allow windowed mode, as configuring parameters requires opening an image in Fiji, for example, and making a few basic measurements. Although I personally use multiple desktops, many users may not be used to this.
- * Numerical indexing of input parameters would enhance user navigation, especially when referring to help in the GUI documentation.
- * Displaying the number of segmented cells per image directly in the GUI after processing would provide users with immediate feedback on segmentation performance.
- * The core Toggle-Untoggle functionality of the tool is great and allows quick elimination of mislabelled cells. Adding the possibility to draw and edit missed or falsely segmented cells would allow further curation of the labels.

2. Data export

- * The tool's use of a tidy, analysis-ready CSV format is great and aligns with best practices for downstream statistical processing.
- * Currently, the tool is designed to process one condition and one replicate at a time. Users would benefit from an option to append results from additional conditions/repeats to an existing CSV file. This would simplify data handling and analysis across experiments.
- * Adding an option to export cell labels in addition to the ROIs would enable faster analysis or visualization in other platforms.
- * When cells are manually untoggled and the output is saved repeatedly, existing CSV and ROI files are overwritten. To avoid data loss, it would be beneficial to add an optional export identifier to allow multiple exports per image. The name could also be derived from experiment/repeat names.

3. Compatibility and Extensibility

- * As discussed in the manuscript, it would be beneficial to expand the tool to be compatible with operating systems beyond Apple Silicon. When done so, implementing GPU-acceleration would significantly reduce processing times.
- * Including support for additional segmentation models, such as Cellpose-SAM, would make the tool more versatile for different biological samples or imaging modalities.
- * Adding the option to use pre-trained models, for example, from model zoos or trained via the Cellpose GUI, would enhance usability.
- * The tool expects two-channel input. Allowing segmentation with only one channel or with brightfield imaging would benefit users working in those contexts.

4. Figures and Presentation

- * All images in the current manuscript are missing scale bars; please ensure these are included.
- * Figures 2 and 3 would benefit from labelling the figure panels.
- * The cell line name MDA-MB-231 should be hyphenated properly.

5. Validation

- * Although the purpose of this tool is not to showcase the performance of the Cyto3 model, it would be beneficial to include benchmarking results (e.g., IoU) comparing the tool's output against manually annotated ground truths. This is especially relevant since Toggle-Untoggle allows adjustment of more parameters than typically required for Cellpose.

* It would be beneficial to demonstrate that Toggle-Untoggle produces segmentation results comparable to those of the Cellpose GUI. If not, due to more parameters (see previous point), please discuss this.

* These points could be addressed in a new figure.

6. Performance and Scalability

* Since the manuscript discusses speed limitations, please include processing time per image or per batch, along with hardware specifications.

* Showcase the tool's scalability by processing larger batches and reporting on time, memory usage, and stability.

* These insights could form a supplementary figure on tool performance.

7. Documentation

* The documentation would benefit from expansion with a demo video, tutorial, and step-by-step walkthrough to enable faster onboarding.

* Please include instructions (with images or video) on how to configure GUI parameters. This is already explained in the GUI but should be part of the documentation as well.

* Provide a direct link to the GitHub repository from within the GUI.

* Clearly list the limitations of the tool in the documentation.

* Include descriptions of the parameters that can be extracted after segmentation.

Reviewer 2

Overview

Grishchenko et al present Toggle-Untoggle, a Graphical User Interface (GUI) for running Cellpose on batches of images and refining the results. Toggle-Untoggle consists of a simple user interface that allows users to modify parameters passed to Cellpose's cyto3 model, which is then run on a directory of images specified by the user. Once the batch run is complete, the interface allows the user to deselect (or, "untoggle") specific cell segmentations produced by Cellpose, such that inaccurate segmentations can be removed from the results.

Intuitive and user-friendly "point and click" softwares can make artificial intelligence (AI) based tools far more accessible to biologists who often lack coding skills and the authors efforts are therefore to be applauded. However, we feel that the utility of Toggle-Untoggle in its current form is somewhat limited and some additional functionality is required in order to make it a truly valuable resource for the community.

Major Limitations

The first major limitation is the lack of an installer for any OS other than Mac. We appreciate why the authors want to make their software easily installable, to avoid users, who may not have any computational skills, having to set up potentially complicated conda environments. But in the absence of an installer for Windows (and Linux), we believe instructions on setting up a conda environment and running from there should be provided. This was in fact the only way we could test the software as we do not have access to Macs!

However, cloning the GitHub repo and setting up a conda environment with the associated YAML file did not work. But, by simplifying the YML file a little and removing restrictions on version numbers, we were able to successfully set up an environment and run the software. The authors should do what they can to make this process easier.

The second major limitation concerns the lack of documentation for the software. There is no documentation at all in the GitHub repo, beyond some very rudimentary installation instructions. There is a Help button (denoted by "?") within the software interface, but the descriptions provided for the various parameters are unlikely to be terribly informative to someone unfamiliar with image segmentation or Cellpose specifically - for example: "16. Actin Channel Intensity Threshold: Adjust to refine segmentation."

Finally, bearing in mind that Cellpose itself ships with a GUI, we feel that the advantages of using Toggle-Untoggle as a separate GUI for Cellpose are somewhat limited. Granted, Toggle-Untoggle allows the batch processing of images. But the main purpose of the software, to allow users to deselect (or "untoggle") undesirable segmentation results, is, we feel, of limited value for a number of reasons:

1. As the authors themselves concede, this does not lend itself to high-throughput analysis. Clicking individual segmentations to remove them is very tedious and time-consuming.
2. As the authors also acknowledge, introducing human interaction into an analysis pipeline runs the risk of introducing subjectivity and bias. It also makes the results of the analysis more difficult to reproduce - two different people looking at the same segmentation results will likely reach different conclusions about which cells should be "untoggled" and which should be retained.
3. Somewhat related to point #2, we would be concerned that this essentially provides people with a tool to "cherry-pick" data. While we appreciate that this is by no means the authors' intention, it should at least be noted in the manuscript.
4. There is no option to load an alternative Cellpose model

We feel that in order for Toggle-Untoggle to truly be valuable to the community, it needs to contain the functionality to allow the fine-tuning of a Cellpose model. Instead of just removing incorrect segmentations, the interface needs to provide the means to correct them. Those corrected segmentations could then form the basis of a new training set to fine-tune a Cellpose model. Ideally, this fine-tuning would be done within the Toggle-Untoggle GUI, outputting a new model which the user could load, test and further refine if needed. We appreciate that including functionality to correct segmentations could make the interface substantially more complex and it is likely the authors' intention to provide users with as simple an interface as possible. But we feel that, in its current form, the functionality is too simple and limited to be of substantial value to the community. Even the ability to split/merge cell masks would be a substantial improvement.

One final concern is the inappropriate use of t-tests in Figures 4 and 5. We refer the authors to the following for guidance on this issue: <https://doi.org/10.1083/jcb.202001064>

Other Specific Comments

p. 7, lines 7-12: 'Upon launch, users select a directory containing input images, which are automatically loaded and processed in batch mode.'

A little more information in the documentation around how input images should be organised would be beneficial. Help provided within the software is very limited.

There is no possibility to minimize the window of the application, which makes it relatively difficult to check for example image properties, while the app is running.

The Pixel-to-Microns ratio field in the app window doesn't appear to have any 'internal control' - it might make sense to limit the range of possible inputs. Alternatively, it should be possible to read this information directly from the input image metadata.

p. 6, lines 38-46: 'All statistical comparisons were conducted using Student's t-test, with a p-value \leq 0.05 considered statistically significant using GraphPad Prism. Data presentation also used GraphPad Prism. Haralick texture features were extracted and analyzed using QuPath software.'

Please specify versions of softwares GraphPad and QuPath used and add relevant references.

Several references are made in the text to the ability to "correct" segmentations, which is not an entirely accurate description of what Toggle-Untoggle does. There is no possibility to 'correct' the segmentation masks within this application, one can only decide whether they include or exclude some cells from analysis.

p. 3, lines 51-54: '(...) a user-friendly correction/verification tool and streamlined extraction of morphological parameters.'

p.4, lines 19-27: 'This interactive correction system provides an efficient way to improve segmentation quality without time-consuming manual editing of masks or identification of outliers in downloaded datasets.'

'verification/correction' p. 7, lines 4 and 19; p. 8, lines 21 and 51

p. 11, lines 4-7: '(...) and do not offer interactive options for correcting segmentation masks, a key feature of "Toggle-Untoggle".'

p. 11, lines 9-14: 'While manual correction of segmentation masks can enhance accuracy, it may introduce some level of subjective bias if not systematically reviewed.'

First revision

Author response to reviewers' comments

POINT BY POINT RESPONSE TO REVIEWER COMMENTS

Reviewer 1:

SUGGESTIONS TO AUTHORS

1. Graphical user interface

* The GUI currently launches in full-screen mode by default. It would be better to allow windowed mode, as configuring parameters requires opening an image in Fiji, for example, and making a few basic measurements. Although I personally use multiple desktops, many users may not be used to this.

* Numerical indexing of input parameters would enhance user navigation, especially when referring to help in the GUI documentation.

* Displaying the number of segmented cells per image directly in the GUI after processing would provide users with immediate feedback on segmentation performance.

* The core Toggle-Untoggle functionality of the tool is great and allows quick elimination of mislabelled cells. Adding the possibility to draw and edit missed or falsely segmented cells would allow further curation of the labels.

RESPONSE: Thank you for your insight on the design of our GUI. We have successfully integrated all of the suggestions mentioned. The GUI now includes a windowed mode, numerical indexing of parameters, an indication of the number of segmented cells per image, as well as additional tools that allow users to merge and draw cell masks.

2. Data export

* The tool's use of a tidy, analysis-ready CSV format is great and aligns with best practices for downstream statistical processing.

* Currently, the tool is designed to process one condition and one replicate at a time. Users would benefit from an option to append results from additional conditions/repeats to an existing CSV file. This would simplify data handling and analysis across experiments.

* Adding an option to export cell labels in addition to the ROIs would enable faster analysis or visualization in other platforms.

* When cells are manually untoggled and the output is saved repeatedly, existing CSV and ROI files are overwritten. To avoid data loss, it would be beneficial to add an optional export identifier to allow multiple exports per image. The name could also be derived from experiment/repeat names.

RESPONSE: We appreciate your comment⁵ and understand that manually appending results from each individual run of the software may be tedious for users without coding expertise. To address this issue and save users time, we have added an additional script called “combine_csvs.py” along with instructions on our GitHub to guide users through merging separate CSV files using the terminal/command line. Regarding the export of cell labels, we considered your suggestion and decided to add labels directly on top of the cells within the GUI. We also revised the ROI naming system from arbitrary to a more structured format: “image_name_label_#”, to improve user navigation. Finally, we ensured that users can generate multiple CSV files and ROI folders by allowing them to rename saved files as many times as desired (via output fields #2 and #3 within the GUI).

3. Compatibility and Extensibility

* As discussed in the manuscript, it would be beneficial to expand the tool to be compatible with operating systems beyond Apple Silicon. When done so, implementing GPU-acceleration would significantly reduce processing times.

* Including support for additional segmentation models, such as Cellpose-SAM, would make the tool more versatile for different biological samples or imaging modalities.

* Adding the option to use pre-trained models, for example, from model zoos or trained via the Cellpose GUI, would enhance usability.

* The tool expects two-channel input. Allowing segmentation with only one channel or with brightfield imaging would benefit users working in those contexts.

RESPONSE: Thank you for your suggestions. In addition to the standalone version of the application for macOS, we created a standalone executable for Windows that is ready to use immediately after download. For Linux users, we added detailed instructions on how to set up an Anaconda virtual environment and run the app within it on our GitHub. We also included the “nuclei” model as one of the default options in the GUI, since the “cyto3” and “nuclei” models are the most commonly used and broadly applicable. In addition, we added an input field that allows users to load their own custom Cellpose model, either trained locally or downloaded from the internet. Unfortunately, we cannot implement the Cellpose-SAM model effectively, as it is extremely computationally expensive and would limit the usability of the software for most users without access to very powerful and costly GPU resources. Finally, we enabled one-channel input, making it possible to segment both single-channel and brightfield images.

4. Figures and Presentation

* All images in the current manuscript are missing scale bars; please ensure these are included.

* Figures 2 and 3 would benefit from labelling the figure panels.

* The cell line name MDA-MB-231 should be hyphenated properly.

RESPONSE: Thank you for these suggestions. We have incorporated these modifications into the new version of our paper.

5. Validation

* Although the purpose of this tool is not to showcase the performance of the Cyto3 model, it would be beneficial to include benchmarking results (e.g., IoU) comparing the tool's output against manually annotated ground truths. This is especially relevant since Toggle-Untoggle allows adjustment of more parameters than typically required for Cellpose.

* It would be beneficial to demonstrate that Toggle-Untoggle produces segmentation results comparable to those of the Cellpose GUI. If not, due to more parameters (see previous point), please discuss this.

* These points could be addressed in a new figure.

RESPONSE: We appreciate these comments and have added a new **Figure 2** to the manuscript comparing the performance of “Toggle-Untoggle” with the “cyto3” model.

6. Performance and Scalability

* Since the manuscript discusses speed limitations, please include processing time per image or per batch, along with hardware specifications.

* Showcase the tool's scalability by processing larger batches and reporting on time, memory usage, and stability.

* These insights could form a supplementary figure on tool performance.

RESPONSE: Thank you for your valuable insight. While we agree that this is an important consideration, our work did not specifically focus on optimizing speed of “Toggle-Untoggle” relative to the Cellpose “cyto3” model. In fact, our implementation is expected to be somewhat slower than “cyto3” when used in a code editor or within the Cellpose GUI, largely due to the added complexity and responsiveness of our GUI design.

7. Documentation

* The documentation would benefit from expansion with a demo video, tutorial, and step-by-step walkthrough to enable faster onboarding.

* Please include instructions (with images or video) on how to configure GUI parameters. This is already explained in the GUI but should be part of the documentation as well.

* Provide a direct link to the GitHub repository from within the GUI.

* Clearly list the limitations of the tool in the documentation.

* Include descriptions of the parameters that can be extracted after segmentation.

RESPONSE: Thank you very much for these suggestions. We have updated our GitHub documentation to better prepare users before they begin working with “Toggle-Untoggle”, including detailed instructions on how to use the application. In addition, we incorporated the GitHub link within the GUI, highlighted the current limitations, and added descriptions of the parameters extracted after segmentation to our documentation.

Reviewer 2:

Major Limitations

The first major limitation is the lack of an installer for any OS other than Mac. We appreciate why the authors want to make their software easily installable, to avoid users, who may not have any computational skills, having to set up potentially complicated conda environments. But in the absence of an installer for Windows (and Linux), we believe instructions on setting up a conda environment and running from there should be provided. This was in fact the only way we could test the software as we do not have access to Macs! However, cloning the GitHub repo and setting up a conda environment with the associated YAML file did not work. But, by simplifying the YML file a little and removing restrictions on version numbers, we were able to successfully set up an environment and run the software. The authors should do what they can to make this process easier.

RESPONSE: We appreciate your feedback. In addition to the macOS version of the standalone application, we have created a standalone executable for Windows. Our documentation now also provides detailed instructions for Linux users on how to set up an Anaconda virtual environment. We extensively tested the contents of our environment.yml file to ensure that the environment can

be reliably set up on any operating system without encountering issues related to package versions or dependencies.

The second major limitation concerns the lack of documentation for the software. There is no documentation at all in the GitHub repo, beyond some very rudimentary installation instructions. There is a Help button (denoted by "?") within the software interface, but the descriptions provided for the various parameters are unlikely to be terribly informative to someone unfamiliar with image segmentation or Cellpose specifically - for example: "16. Actin Channel Intensity Threshold: Adjust to refine segmentation."

RESPONSE: Thank you for your comment. We have enhanced our documentation by providing detailed installation instructions, explanations of each input parameter, descriptions of the parameters that can be extracted, and troubleshooting tips. Additionally, we included guidelines for users who wish to extend their analysis beyond the platform, such as quantifying Haralick texture features for cells segmented using "Toggle-Untoggle" via one of the scripts provided on our GitHub.

Finally, bearing in mind that Cellpose itself ships with a GUI, we feel that the advantages of using Toggle-Untoggle as a separate GUI for Cellpose are somewhat limited. Granted, Toggle-Untoggle allows the batch processing of images. But the main purpose of the software, to allow users to deselect (or "untoggle") undesirable segmentation results, is, we feel, of limited value for a number of reasons:

1. As the authors themselves concede, this does not lend itself to high-throughput analysis. Clicking individual segmentations to remove them is very tedious and time-consuming.
2. As the authors also acknowledge, introducing human interaction into an analysis pipeline runs the risk of introducing subjectivity and bias. It also makes the results of the analysis more difficult to reproduce - two different people looking at the same segmentation results will likely reach different conclusions about which cells should be "untoggled" and which should be retained.
3. Somewhat related to point #2, we would be concerned that this essentially provides people with a tool to "cherry-pick" data. While we appreciate that this is by no means the authors' intention, it should at least be noted in the manuscript.
4. There is no option to load an alternative Cellpose model

RESPONSE: We appreciate your insights. While we cannot completely eliminate the need for user intervention in our application, since allowing users to interact with the analysis on small datasets was deliberately intended to help wet-lab researchers improve the accuracy of their segmentation results with minimal manual effort, we have highlighted all of the potential drawbacks you mentioned in the limitations section of the manuscript and the GitHub documentation. Additionally, we have added an option for users to load a custom Cellpose model, either trained locally or downloaded from the internet.

We feel that in order for Toggle-Untoggle to truly be valuable to the community, it needs to contain the functionality to allow the fine-tuning of a Cellpose model. Instead of just removing incorrect segmentations, the interface needs to provide the means to correct them. Those corrected segmentations could then form the basis of a new training set to fine-tune a Cellpose model. Ideally, this fine-tuning would be done within the Toggle-Untoggle GUI, outputting a new model which the user could load, test and further refine if needed. We appreciate that including functionality to correct segmentations could make the interface substantially more complex and it is likely the authors' intention to provide users with as simple an interface as possible. But we feel that, in its current form, the functionality is too simple and limited to be of substantial value to the community. Even the ability to split/merge cell masks would be a substantial improvement.

RESPONSE: Thank you for your insights. To address your concern while maintaining a user-friendly interface, we have added tools to the GUI that allow users to merge neighboring masks and draw new ones, which can then be seamlessly incorporated into the rest of the analysis.

One final concern is the inappropriate use t-tests in Figures 4 and 5. We refer the authors to the following for guidance on this issue: <https://doi.org/10.1083/jcb.202001064>

RESPONSE: We appreciate your concerns and have revised the approach to our statistical analysis in the latest version of the manuscript.

Other Specific Comments

p. 7, lines 7-12: 'Upon launch, users select a directory containing input images, which are automatically loaded and processed in batch mode.'

A little more information in the documentation around how input images should be organised would be beneficial. Help provided within the software is very limited.

RESPONSE: Thank you for your feedback. We have added guidelines to our documentation explaining how users should organize their images before using the software.

There is no possibility to minimize the window of the application, which makes it relatively difficult to check for example image properties, while the app is running.

RESPONSE: Thank you for your comment. We have ensured that the GUI now includes a windowed mode, allowing users to access other windows while "Toggle-Untoggle" is running.

The Pixel-to-Microns ratio field in the app window doesn't appear to have any 'internal control' - it might make sense to limit the range of possible inputs. Alternatively, it should be possible to read this information directly from the input image metadata.

RESPONSE: We appreciate your suggestion. We added a value constraint to the field for the pixel-to-micron ratio, which can now only be set between 0 and 2.

p. 6, lines 38-46: 'All statistical comparisons were conducted using Student's t-test, with a p-value ≤ 0.05 considered statistically significant using GraphPad Prism. Data presentation also used GraphPad Prism. Haralick texture features were extracted and analyzed using QuPath software.'

Please specify versions of software GraphPad and QuPath used and add relevant references.

RESPONSE: Thank you for your comment. We have now included the versions of all software used in the Methods section.

Several references are made in the text to the ability to "correct" segmentations, which is not an entirely accurate description of what Toggle-Untoggle does. There is no possibility to 'correct' the segmentation masks within this application, one can only decide whether they include or exclude some cells from analysis.

p. 3, lines 51-54: '(...) a user-friendly correction/verification tool and streamlined extraction of morphological parameters.'

p.4, lines 19-27: 'This interactive correction system provides an efficient way to improve segmentation quality without time-consuming manual editing of masks or identification of outliers in downloaded datasets.'

'verification/correction' p. 7, lines 4 and 19; p. 8, lines 21 and 51

p. 11. lines 4-7: '(...) and do not offer interactive options for correcting segmentation masks, a key feature of "Toggle-Untoggle".'

p. 11, lines 9-14: 'While manual correction of segmentation masks can enhance accuracy, it may introduce some level of subjective bias if not systematically reviewed.'

RESPONSE: We appreciate your concern and have added features such as merging cell masks and drawing new cell outlines to our GUI, ensuring that “Toggle-Untoggle” can now be effectively used as a tool for correction and verification. We also acknowledge that “correct/correction” is not an accurate description of what “Toggle-Untoggle” was intended to do, and this has been corrected in the manuscript.

Second decision letter

MS ID#: jcs.264154R1

MS TITLE: Toggle-Untoggle: A cell segmentation tool with an interactive user verification interface

AUTHORS: Nina Grishchenko; Margarita Byrsan; Matthew Craig Drummond-Stoyles; Michael F Olson

ARTICLE TYPE: Tools and Resources

Dear Dr Olson,

We have now reached a decision on the above manuscript.

To see the reviewers' reports and a copy of this decision letter, please go to:

As you will see, the reviewers gave favourable reports but raised some critical points that will require amendments to your manuscript. I hope that you will be able to carry these out because I would like to be able to accept your paper, depending on further comments from reviewers.

Reviewer 1

I would like to thank the authors for carefully addressing my comments. The revised version shows clear improvements in both software usability and showcasing example cases. Significant effort has been made to improve the "Toggle-untoggle" tool, documentation, and the manuscript.

While the manuscript is now much improved, I would like to offer two further recommendations that I believe could be beneficial:

1. Thank you for improving the CSV export part. It was very easy to change the name of the experiment and export, thus avoiding overwriting of the previously exported results. However, from the user interface this was not obvious. I would recommend including a small instruction on how to avoid overwriting near the CSV export button in the GUI.
2. I would add another title at the end of the manuscript called "Code availability" and include the link to GitHub. This will help the reader quickly find the repository and start testing the tool without having to search through the manuscript for the link.

Overall, the manuscript and the tool have benefited substantially from the revisions. With these additional refinements, I am happy to recommend this manuscript for publication.

Reviewer 2

I appreciate the authors' efforts in addressing the points I raised previously and thank them for their point-by-point response. The updates to the Github repo and the application mean they are much improved. I have however just two further comments on the revised application:

* Users are required to type/copy-paste into the Images Folder Path field in the GUI - a folder selection dialogue here might make the process a little more straightforward?

* I've just run the application on a test image with approximately 100 cells and the Images in the resultant annotations under the Processed Images tab are quite small - perhaps a Zoom feature would help here to aid with inspection of the results?

Second revision

Author response to reviewers' comments

Response to reviewers' comments

Reviewer 1: I would like to thank the authors for carefully addressing my comments. The revised version shows clear improvements in both software usability and showcasing example cases. Significant effort has been made to improve the "Toggle-untoggle" tool, documentation, and the manuscript.

While the manuscript is now much improved, I would like to offer two further recommendations that I believe could be beneficial:

1. Thank you for improving the CSV export part. It was very easy to change the name of the experiment and export, thus avoiding overwriting of the previously exported results. However, from the user interface this was not obvious. I would recommend including a small instruction on how to avoid overwriting near the CSV export button in the GUI.

RESPONSE: Thank you for pointing out that it might be helpful to inform users about the possibility of overwriting files they may still need. We have added comments on how to avoid overwriting both the .csv file and the ROI folder in our GitHub documentation as well as in the instruction window within the GUI.

2. I would add another title at the end of the manuscript called "Code availability" and include the link to GitHub. This will help the reader quickly find the repository and start testing the tool without having to search through the manuscript for the link.

RESPONSE: Thank you for this helpful suggestion. This new section has now been incorporated into the paper.

Overall, the manuscript and the tool have benefited substantially from the revisions. With these additional refinements, I am happy to recommend this manuscript for publication.

Reviewer 2: I appreciate the authors' efforts in addressing the points I raised previously and thank them for their point-by-point response. The updates to the GitHub repo and the application mean they are much improved. I have however just two further comments on the revised application:

* Users are required to type/copy-paste into the Images Folder Path field in the GUI - a folder selection dialogue here might make the process a little more straightforward?

RESPONSE: Thank you for noting that. We completely agree that this would simplify the folder lookup process and have added a browse button within the image path selection section, allowing users to easily retrieve the path to the necessary folder without leaving the GUI.

* I've just run the application on a test image with approximately 100 cells and the Images in the resultant annotations under the Processed Images tab are quite small - perhaps a Zoom feature would help here to aid with inspection of the results?

RESPONSE: Thank you for this suggestion. We have added a feature that allows users to zoom and pan all images displayed in the second tab of the GUI. It definitely makes inspection of the results easier.

Third decision letter

MS ID#: jcs.264154R2

MS Title: Toggle-Untoggle: A cell segmentation tool with an interactive user verification interface

Authors: Nina Grishchenko; Margarita Byrsan; Matthew Craig Drummond-Stoyles; Michael F Olson

Article Type: Tools and Resources

Dear Dr Olson,

I am happy to tell you that your manuscript has been accepted for publication in Journal of Cell Science, pending standard publication integrity checks.